# Growth, Productivity, Yield Components and Seasonality of Different Genotypes of Forage Clover *Lotus corniculatus* L. under Varied Soil Moisture Contents

**DOI:** 10.3390/plants13101407

**Published:** 2024-05-18

**Authors:** Sahara Xolocotzi-Acoltzi, Aurelio Pedroza-Sandoval, Gabino García-De los Santos, Perpetuo Álvarez-Vázquez, Isaac Gramillo-Ávila

**Affiliations:** 1Unidad Regional Universitaria de Zonas Áridas de la Universidad Autónoma Chapingo, Km. 40 Carretera Gómez Palacio—Ciudad Juárez, Bermejillo C.P. 35230, Durango, Mexico; xolocotzi18@gmail.com (S.X.-A.); igramillo4@gmail.com (I.G.-Á.); 2Departamento de Recursos Naturales del Colegio de Posgraduados, Campus Montecillo, Km. 36.5 Carretera México-Texcoco, Montecillo, Texcoco C.P. 56230, Estado de México, Mexico; garciag@colpos.mx; 3Departamento de Recursos Naturales de la Universidad Autónoma Agraria Antonio Narro, Calzada Antonio Narro 1923, Buenavista, Saltillo C.P. 25315, Coahuila, Mexico; perpetuo.alvarezv@uaaan.edu.mx

**Keywords:** drought, stress physiology, agronomic indicators, plant stress

## Abstract

This study aimed to evaluate the response to water deficit of different ecotypes and a variety of *Lotus corniculatus* on growth, productivity, and yield components, through seasonal times. A randomized block experimental design in a 2 × 5 factorial arrangement with three replicates was used. The first variation factor was soil moisture contents: field capacity (FC) was 26.5% ± 1.5, and water deficit (WD) was 85% of FC (22.5% ± 1.5); the second variation factor was the ecotypes identified with the codes 255301, 255305, 202700 and 226792 and of the variety Estanzuela Ganador. The best responses in plant cover and weight of accumulated fresh biomass were obtained in the ecotype 202700 under WD, with values of 1649.0 cm^2^ and 583.7 g plant^−1^, and 1661.2 cm^2^ and 740.1 g plant^−1^ in ecotype 255305 under FC. The leaf clover was the main component of yield during the summer and autumn seasons. Ecotype 226792 was tolerant to low temperatures during the winter season with better leaf development. Ecotype 202700 is the best option for forage clover production when water is limited, and ecotype 255305 when water is not resource-limited, but these preliminary conclusions need to be confirmed in field studies.

## 1. Introduction

Climate change and environmental impact are some of the causes of the degradation of natural resources, in addition to anthropogenic action, which compromises the stability of agroecosystems and the consequent agri-food productivity given the increase in the world population [1]. Agriculture is the productive activity with the greatest demand for resources and inputs, especially water resources, with a consumption of 85% of the water available used for agri-food production [2].

Water scarcity is the most recurrent problem due to the lack of rain with increasingly intense and frequent droughts in the world [3]. Due to droughts, bodies of water are reduced, and the recharge of aquifers is poor due to the overexploitation of water resources in irrigation areas with deep well water, with negative effects of low quantity and quality of the water for various domestic and productive uses [4,5].

This imbalance between the volume of water extracted and the increasingly diminished recharge of aquifers already has serious economic, social, and environmental impacts in different forage-producing regions [6,7,8]. An alternative to the problem of forage monoculture with high water requirements in areas with low availability of quality water is the diversification of crops with the same forage type, with productive potential but with lower water demand [9,10,11]. Biodiversity through the introduction of alternative forages holds promise in mitigating these challenges due to their drought tolerance and adaptability to various environmental conditions [12,13,14]. In contexts such as those observed by this study, livestock producing milk and meat on a large scale are the most demanding of forage for feed under stable conditions [15]. Alfalfa (*Medicago sativa* L.) is a forage crop that is excellent at producing great quantities of fresh matter with high nutritional content to produce milk and meat; however, this species is a semi-perennial crop requiring irrigation throughout the year. Alfalfa is a plant with the greatest water demand for growing and development, which makes it a crop species with low efficiency in water use [7,16].

The diversification of forage crops with productive potential but with lower water demand is an alternative to the problem of monoculture with high water requirements in areas with low availability of water [10]. Some alternatives are native crops such as nopal [17,18] or introduced crops such as forage clover [19,20] which are viable alternatives for irrigated agricultural areas affected by the problem of water scarcity.

Recently, the performance of legume species has taken on special relevance, which have diverse adaptation properties to marginal conditions and contribute to sustainable agriculture and agri-food systems [11,21]. Different forage species of clover are being explored through research and technological development programs that can compete with crops such as alfalfa to maintain productive quality and quantity with more efficient use of water [22]; bird’s foot clover *Lotus corniculatus* L. is the most outstanding of these. This forage species is a forage legume of Mediterranean European and African origin, which has a high capacity to adapt to different environments, such as water deficit, nutritional deficiencies, low temperatures, and tolerance to soils with acidic pH, as well as to high levels of aluminum and manganese [23]. This species has a productive performance comparable to alfalfa or forage clover *Trifolium repens* L. [24]. The growth of *L. corniculatus* is successful in countries with dry summers and marked seasonality, such as New Zealand and Uruguay, where it has shown good response behavior as a forage crop [25] and tolerance to water deficit by drought [26].

Bird’s-foot trefoil, a prevalent grass-legume, thrives in poorly drained acid soils and demonstrates adaptability to sandy soils [27,28]. Unlike other forage legumes in eastern North America, it does not induce bloat in ruminants [29]. The species has three distinct growth habit types—prostrate, erect, and semi-erect—with each type catering to specific agricultural purposes. Seedling vigor is comparatively lower, and the species utilizes root reserves for early spring growth, unlike most legumes that replenish root reserves in late summer [27]. However, the widespread establishment of this crop has not been easy to solve since the complexity of its reproduction by seed and its relatively slow growth make it difficult to expand it in intensive production systems [30]. This study aimed to evaluate different genotypes of *L. corniculatus* in their capacity to respond to growth, production, and yield under water deficit through seasonal times.

## 2. Results

### 2.1. Climatic Conditions within the Shade Mesh

According to the temperature report during the study time, an average maximum of 42.3 °C and an average minimum of 7.7 °C were identified, with a maximum of 46.9 °C and a minimum of −4.6 °C. The average relative humidity recorded ranged between 44 and 73%, with a minimum between 5 and 10% in the months from May to July and a maximum of 100% in the rainy periods distributed across the months July, August, and September; both variables were measured during the period from April 2021 to May 2022 (Figure 1). 

### 2.2. Stem Length

The length of stems did not vary significantly (*p* ≤ 0.05) in soil moisture content at FC in the different genetic materials of *L. corniculatus* that were evaluated, while in the WD treatments, a genotypic variation was recorded with better behavior of ecotype 202700, with a length of 23 cm, compared to ecotypes 255305, 226792 and the Estanzuela Ganador variety, which recorded the lowest growth with values of 19.5, 19.1 and 19.7 cm, respectively (Table 1).

### 2.3. Plant Coverage

Plant cover was significantly greater (*p* ≤ 0.05) in ecotypes 255305 and 202700 under FC and WD, with values of 1661.2 and 1649 cm^2^, respectively, compared to ecotypes 226792 and 255301, which showed the lowest values of 1249 and 1299 cm^2^ in FC, respectively. The ecotype 226792 and the Estanzuela Ganador variety had the lowest values under WD, with values of 1121.5 and 1117.1 cm^2^, respectively (Table 1).

### 2.4. Production of Accumulated Fresh Biomass (PAFB)

The PAFB of different Lotus genotypes was like that of plant coverage (*p* ≤ 0.05) due to the relationship between both variables; ecotypes 255305 and 202700 were the ones with the best response, with values of 740.1 and 583.7 g plant^−1^ under FC and WD, respectively. Ecotype 255301 was the one with the lowest response, with 410.6 g plant^−1^. The rest of the genetic materials had an intermediate behavior across both response effects (Table 1).

### 2.5. Performance Components

The contribution of the components to the forage yield of the genetic materials evaluated showed variability depending on the season but was similar in both soil moisture contents FC and WD. The range of contribution of leaves was 50–80%, stems 15–40%, and dry stems 1–10%. Inflorescences showed percentages of 0.5–4%, but only in 202700 and Estanzuela Ganador in the periods from Spring to Summer (SP-SU) under FC, and SU under both FC and WD.

The winter was the period with the lowest contribution of leaves to the yield, with a range of 46–60%, except for ecotype 226792, which maintained a contribution of 60–70% under FC and WD. Ecotypes 255301 and 226792 registered the highest values of leaf contribution with 60–80%, stems 15 to 25%, and dry stems 0–5% on the day of cutting; ecotype 202700 recorded the lowest values in this variable, with 45–55%, stems 20–45%, dry stems 0–30%, and inflorescences 0–4% (Figure 2).

### 2.6. Temporal Dynamics of Plant Growth

The variables stem length, plant cover, and accumulated fresh biomass did not have a significant difference (*p* ≤ 0.05) between genotypes, but they did differ according to the harvest period. The highest values of stem length corresponded to the Summer–Winter (SU-W) periods, where lengths of 25.7 cm were observed under WD and 29.1 cm under FC. The vegetation cover on the day of cutting in the SU-W period observed very similar behavior in both moisture contents, with values of 2363.2 cm^2^ and 2300.7 cm^2^ under FC and WD, respectively (Figure 3).

The lowest values of this variable occurred in the SP-SU periods, with 443.2 and 396.2 cm^2^, and W, with 992.8 and 960.6 cm^2^, in both soil moisture contents FC and WD, respectively. In terms of PAFB at 85% CC, the Summer–Autumn (SU-A) and SP times were the most productive, and the winter period had the lowest productivity; however, in W-SP and SP, there was a similar production in both soil moisture contents (Figure 3).

## 3. Discussion

According to Tyree et al. [31] the first stages of plant development are critical, as in the results obtained in our study, since the materials evaluated showed a pattern of slow growth and low productivity in the first months of evaluation, tolerating abiotic stress factors, mainly thermal with the presence of high temperatures (46.9 °C) and low temperatures (−4.6 °C). These results coincide with those reported by García-Bonilla et al. [32] when evaluating different genetic materials of *L. corniculatus*, but in temperate climates, where the highest values of plant height (23–24 cm) were recorded in periods with temperatures above 30 °C, and less growth (10 cm) in the winter season, with temperatures of 0–14 °C. Álvarez-Vázquez et al. [33] point out that the optimal growth temperature for Lotus is 22 °C, which corresponds to average temperatures that occur in most of the year in the region of this study.

The low response effect on *L. corniculatus* growth, measured in terms of stem length, indicates that, although the plants are different in growth habit, all ecotypes/varieties develop at similar growth rates under an optimal regime of soil moisture, but with variation between genotypes under deficient irrigation. That means that the genetic materials identified in this study differ in their tolerance to water deficit, which allows us the opportunity to select the best phytogeneric materials to tolerate water stress in plants in regions with dry climates [34]. Plant covering was favored by the optimal irrigation condition (FC) in the ecotype 255305, which stood out, while 202700 was the most tolerant to WD (85% FC).

The best effect in PAFB in ecotype 202700 under water-deficient conditions suggests the good performance of the plant’s metabolism under these restrictive conditions of soil moisture content. Tolerance to water deficit depends largely on the characteristics of the species, related to the ability to optimize carbon assimilation processes and moderate levels of transpiration [35]. Reports from IICA [26] show that *L. corniculatus* registered the highest values of stomatal conductance and the highest water potentials under water stress, which suggests that this species increases efficiency in water use under water deficit.

The best response in terms of yield components is a high leaf, which is the main forage component of the evaluated plants, and the ecotypes 255301 and 226792 stood out the most in this regard. This effect was maintained in terms of plant covering, but not in the production of accumulated fresh biomass. The importance of leaf production in forages greater than 50% lies in the fact that it is the most active organ of metabolism in the plant, and consequently has the highest concentration of easily degradable metabolites [36], proteins, and other nitrogenous compounds [37].

Regarding PAFB, the ecotypes with the best response were 255305 and 202700 under FC and WD, respectively; however, 202700 under water deficit had a high percentage of stem (>35%) as a component of the forage yield, which is considered an undesirable characteristic since stems are relatively rigid and contain more lignified tissues that are less digestible [36].

The results obtained coincide with those reported by Álvarez-Vázquez et al. [38], whereby the productive evaluation of ectype 202700 showed that the leaf component was 53% of the annual yield, followed by stems with 32%, dry steams 8%, and weeds 7%. Similar values were reported by the same authors in another study, where leaves contributed 56%, stems 30.5%, dry steams 8.5%, and weeds 4.5% [33]. In the case of our study, the presence of dead material was greater in the periods W, W-SP, and SU, with low-temperature stress being the one that notably affected the productivity of clover *L. corniculatus*. However, ecotype 226792 was identified as the most tolerant to this type of stress, regardless of the soil moisture content established in this study.

The seasonal effects on forage growth and the variation of the components of its yield are influenced by environmental conditions such as temperature variation rather than the moisture content in the soil, which coincides with Festo et al. [39] who reported that plants increase leaf coverage in temperatures between 20 and 35.2 °C, but decrease when they exceed 35 °C.

In the results obtained, a significant contribution of dry stems of up to 10% was observed on the day of cutting in the winter and spring periods, which could affect the quality of the forage, since the age of the plant is a determining factor in the distribution of dry matter in the tissues; with increasing plant age, the proportion of stems and senescent material increases, and leaf formation decreases [40]. Therefore, the growing periods of *L. corniculatus* are reduced to less than 45 days in summer and autumn and less than 90 days in winter.

## 4. Materials and Methods

### 4.1. Geographic Location of the Study Area

The experiment was carried out at the Unidad Regional Universitaria de Zonas Áridas of the Universidad Autónoma Chapingo in Bermejillo, Durango, Mexico, located at 25.8° LN and 103.6° LW at an altitude of 1130 m. The area corresponds to a desert climate, with rain in summer and cool winters, an average annual rainfall of 258 mm, an average annual potential evaporation of 2000 mm, and an average annual temperature of 21 °C, with a maximum of 33.7 °C and a minimum of 7.5 °C [41].

### 4.2. Preparing the Experiment

In the experiment, rigid plastic pots of 20 kg capacity with dimensions of 35 cm in diameter and 31.3 cm in height were used, each one containing 18 kg of a substrate mixture: 50% soil, 30% compost, and 20% sand (Figure 4). The physical and chemical characteristics were 22% silt, pH of 8.69, electric conductance (EC) of 10.76 dS m^−1^, and apparent density of 1.46 g cm^−3^.

Transplanting each ecotype/variety into a pot was completed using a seedling rhizome, which was previously grown in a black plastic bag of 1 kg capacity for 90 days; after this time, the transplant was carried out into the pot when the rhizome of the seedling had sufficient growth, development, and vigor. For 62 days after transplanting, irrigation was maintained at FC, and subsequently, a standardization foliage cut was made at a height of 6 cm from the base of the substrate.

For irrigation of the clover plants in each pot, the soil moisture content was monitored in real time with the use of the Model HB-2 digital tensiometer equipment (Ontario, Canada). Once the soil moisture content treatments were differentiated (FC, and 85% FC), the moisture content of each treatment decreased to a minimum level corresponding to 25% and 21% in FC and WD conditions, respectively, and water levels were manually raised to their maximum levels, corresponding to 28% (0.5% more than FC) and 24% in the WD condition, applying 0.54 L for 18 kg of soil per pot for each period of irrigation to raise the maximum level of each pot’s soil moisture content. This volume of water was applied at each irrigation time, varying the interval time according to the evaporation rate of the season. A margin of 3.5% of the usable moisture range (17.5–21%) was left to prevent reaching PWP.

### 4.3. Experimental Design

A randomized block experimental design in split plot arrangement was used with three replicates, under shade mesh conditions. This experimental design was used for practical reasons, where each row of pots, named large plots, corresponded to a soil moisture content in an experimental field, and within each row (large plot), the five Lotus ecotypes were placed. Then, the large plots were assigned two soil moisture contents: field capacity (FC), maintaining an interval of soil moisture content of 26.5% ± 1.5 (25–28%), and water deficit (WD: 85% FC), 22.5% ± 1.5 (21–24%), according to the soil moisture properties obtained using the pot method membrane [42], where FC was 27.5% and the permanent wilting point (PWP) was 17.5% (Figure 5). For the optimal level of soil moisture, the field capacity was maintained between the range of 25 and 28%, where the upper limit is slightly higher than FC, only 0.5% more, due to ongoing evaporation. This is not a significant excess, and nor does it imply a water leak, since it soon dropped to its lower limit (25%), almost always remaining at an average level of 26.5%, which corresponds to the FC. Small plots were four ecotypes and one variety of *Lotus corniculatus*, with IDs 255301, 255305, 202700, 226792, and Estanzuela Ganador variety (Table 2).

An ORIA brand digital thermometer–hygrometer (Model OUS-WA62, China) was placed inside the shade mesh, with which the daily maximum and minimum temperatures and relative humidity were recorded during the study period from March 2021 to May 2022.

The experiment was covered with a plastic roof during the rainy period to avoid alterations in the soil moisture content. The measurements of the respective variables began in April 2021, and a total of seven harvest cuts of fresh foliage (forage) were made across the seasons of the year and the periods between them: spring–summer, summer, summer–autumn, autumn, winter, winter–spring, and spring. The first cut was carried out on 20 July 2021, and the last one on 12 May 2022. No fertilizer application was carried out in the experiment.

### 4.4. Measured Variables

#### 4.4.1. Stem Length

Stem length (cm) was measured with a graduated ruler from the basal part of the plant for both the erect and prostrate ecotypes/variety.

#### 4.4.2. Plant Coverage

Plant coverage (cm^2^), which is an area obtained from the diameters of the foliage crown measured from the center of the bud with a tape measure, was obtained using the following equation [43]:$$PC=\frac{\pi *d_{1}*d_{2}}{4}$$
where PC = plant cover (cm^2^); d_1_ = diameter in the north–south direction (cm); and d_2_ = diameter in the east–west direction.

#### 4.4.3. Production Accumulated Fresh Biomass

The production of accumulated fresh biomass (PAFB) (g plant^−1^) was quantified by making foliage cuts from a height of 6 cm above the ground on each sampling date, and the accumulated production was obtained by summing all the cuts made during the year.

#### 4.4.4. Yield Components of Forage

Yield components of forage refer to the proportion of each morphological component of the plant in its contribution to biomass production, for which 10 stems were taken at random per treatment, separating the inflorescences, dry stems, leaves, and stem, each part used as subsamples of the total return component. The subsamples were dried in a HAFO^®^ Brand forced air oven (Model 1600, SHEL-LAB, Cornelius, OR, USA) at a temperature of 60 °C for 24 h or until constant weight. After this time, the partial and total dry weight were recorded to determine the percentage contribution to yield, for which the following equation was used [33]:$$PMC=\left[\frac{SC}{TDM}\right]\times 100$$
where PMC = percentage per morphological component (%), SC = subsample of the component (g of dry matter), and TDM = total dry matter (g dry matter).

### 4.5. Analysis of Data

The database was processed using Statistical Analysis System Software [44] version 9.0, with which an analysis of variance and multiple range test of Tukey means (*p* ≤ 0.05) were performed to identify the treatment effect. Additionally, Excel program version 6.0 was used for regression analysis.

## 5. Conclusions

The best response in plant cover and weight of accumulated fresh biomass with values of 1649.0 cm^2^ and 583.7 g plant^−1^ was obtained in the 202700 ecotypes under water deficit and 1661.2 cm^2^ and 740.1 g plant^−1^ in ecotype 255305 under optimum soil moisture content. The leaf clover was the main component of yield during the summer and autumn seasons. Ecotype 226792 was tolerant to low temperatures during the winter season, with better leaf development. Thus, ecotype 202700 is the best option for forage clover production when water is limited, and ecotype 255305 when water is not resource-limited, but these preliminary conclusions need to be confirmed in field studies

## Figures and Tables

**Figure 1 plants-13-01407-f001:**
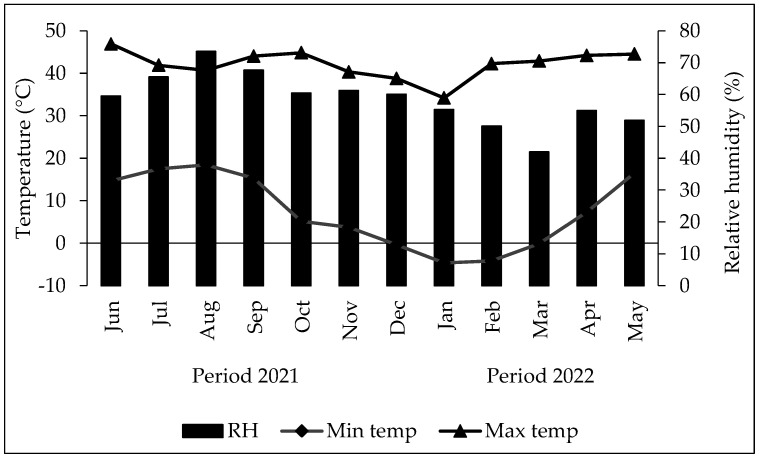
Variation in temperature (°C), and relative humidity (%) within the shadow mesh of the experimental area from June 2021 to May 2022.

**Figure 2 plants-13-01407-f002:**
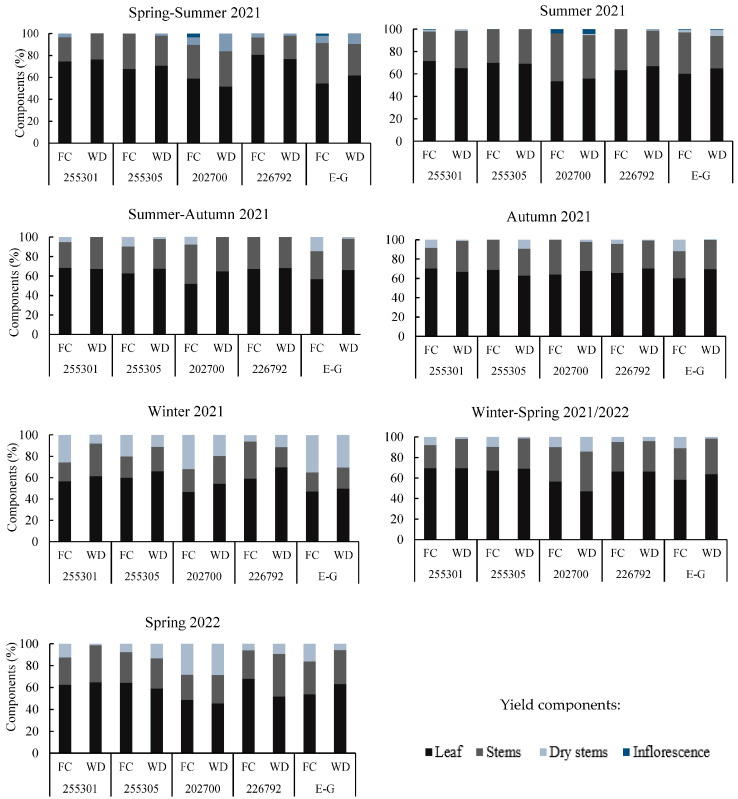
Performance components of different genetic materials of *L. corniculatus* according to season at field capacity (FC) and water deficit (WD: 85% FC).

**Figure 3 plants-13-01407-f003:**
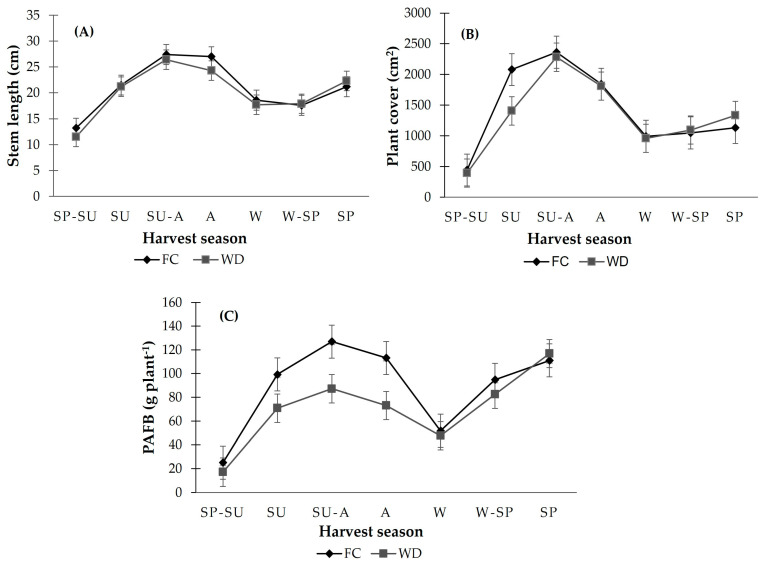
Temporal dynamics of stem length (**A**), plant cover, and (**B**) accumulated fresh biomass production (PAFB) (**C**) of *L. corniculatus* at different seasonal times of the year in optimal content (CF) and water deficit (WD: 85% FC) during the period from April 2021 to May 2022. Spring–Summer (S-SU), Spring (SP), Summer (SU), Summer–Autumn (SU-A), Autumn (A), Winter (W), and Winter–Spring (W-SP) correspond to periods of harvest seasons.

**Figure 4 plants-13-01407-f004:**
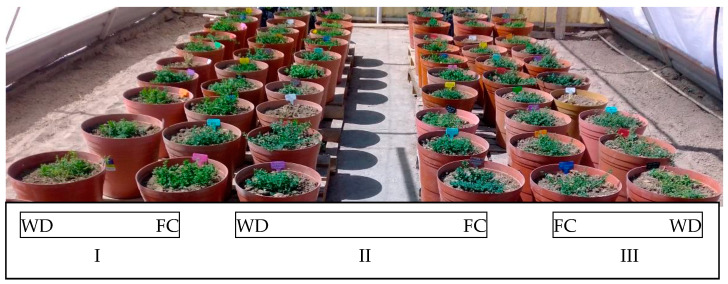
General view of the experiment of different ecotypes and a variety of *Lotus corniculatus* L. at different soil moisture contents under mesh-shade conditions. WD is water deficit; FC is field capacity; I, II, and III are the replicates. Each pot row is a large plot, and each pot is an ecotype/variety of *L. corniculatus* L.

**Figure 5 plants-13-01407-f005:**
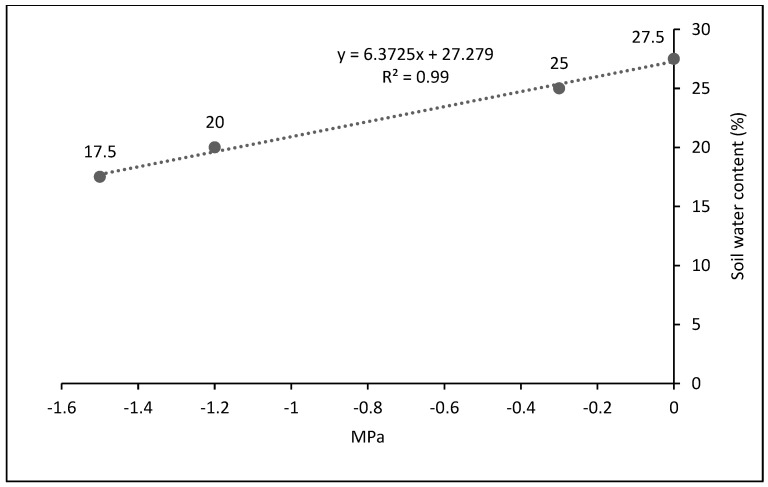
Soil moisture abatement curve [42] of the substrate used in the pots containing *Lotus corniculatus* L. under shade mesh conditions.

**Table 1 plants-13-01407-t001:** Growth, development, and forage productivity of different ecotypes and varieties of *L. corniculatus* under two soil moisture contents.

Ecotypes/Variety	Stem Length (cm)		Plant Coverage (cm^2^)		PAFB (g plant^−1^)	
	FC	WD	FC	WD	FC	WD
255301	20.5 a ± 6.0	21.6 ab ± 7.2	1299.7 b ± 643.7	1252.8 ab ± 880.1	605.8 ab ± 54.5	410.6 b ± 118.7
255305	22.0 a ± 5.2	19.5 b ± 4.8	1661.2 a ± 875.5	1244.3 ab ± 681.4	740.1 a ± 58.9	497.6 ab ± 94.6
202700	21.8 a ± 7.0	23.0 a ± 4.3	1530.0 ab ± 920.2	1649.0 a ± 756.5	571.3 b ± 160.9	583.7 a ± 45.7
226792	20.3 a ± 5.8	19.1 b ± 5.3	1249.0 b ± 589.1	1121.5 b ± 597.7	554.7 b ± 322.2	497.7 ab ± 278.5
Estanzuela Ganador	21.2 a ± 4.6	19.7 b ± 4.4	1334.1 ab ± 671.8	1117.1 b ± 498.9	640.4 ab ± 116.9	486.5 ab ± 283.0

Tukey test (*p* ≤ 0.05). Figures with the same letters within the same column are statistically equal. FC is field capacity (26.5% ± 1.5); WD is water deficit at 85% FC (22.5% ± 1.5); PAFB is the production of accumulated fresh biomass; values after ± are standard deviations.

**Table 2 plants-13-01407-t002:** The relationship, origin, and growth habit of *Lotus corniculatus* ecotypes were evaluated in the experiment.

Ecotypes/Variety	Country of Origin	Growth Habit
255301	Francia	Semi erecto
255305	Italia	Semi erecto
202700	Uruguay	Erecto
226792	Canadá	Semi erecto
Estanzuela Ganador	Uruguay	Erecto

## Data Availability

Data are available on request to the corresponding author’s email with appropriate justification.

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
