# Peer review of "Growth, Productivity, Yield Components and Seasonality of Different Genotypes of Forage Clover Lotus corniculatus L. under Varied Soil Moisture Contents"

_plants, 2024, doi:10.3390/plants13101407_

Round 1
Reviewer 1 Report
Comments and Suggestions for Authors
In this study, a pot experiment was conducted to examine the seasonal growth and yield of forage clover under different soil moisture and genotypes. The aim was to identify the most productive forage clover varieties and the optimal seasons for yield contribution under different soil moisture conditions in Mexico. The findings offer practical insights for the scientific management of water resources to enhance forage clover production. However, several areas within the paper require refinement. Here are the specific suggestions for improvement:
1. Title: Given that the paper concluded with the identification of suitable forage clover varieties under two soil moisture treatments, it is recommended to revise the title to "Growth, productivity, yield components, and seasonality of different genotypes of forage clover Lotus corniculatus L. under varied soil water conditions."
2. Abstract: It is suggested to simplify the labels of different genotypes for forage clover, the current labels are too long to distinguish. Additionally, the abstract should include a concise summary of the study's conclusions. Also, I don't know that the data belong to which ecotype in Line 23, or the average value of the two ecotypes?
3. The introduction section overemphasizes the development and utilization of water resources in Mexico and the significance of pasture cultivation, simplification should be done. There is insufficient discussion on the research status and challenges related to the paper's focus, which is the impact of soil moisture on forage clover growth and yield. This oversight diminishes the discussion on the study's rationale and importance, making the paper read more like a report than a research.
4. Results: The study is designed with two factors, soil moisture and ecotype of forage clover, with two levels for soil moisture and five for ecotype, constituting a two-factor experiment. However, the ANOVA presented focuses on one-way ANOVA by ecotype, whereas a two-way ANOVA should be applied to discern the significance of differences across various soil moisture levels and the interaction between soil moisture and ecotype. Moreover, the results should concentrate on the relative variation among treatments rather than absolute values, which are less meaningful.
5. Line 101-103: The author's intent is unclear. It is mentioned that, as showed in Figure 1, the average minimum temperature did not reach 20°C, while the average maximum temperature exceeded 30 °C, why?
6. Figure 3: It is suggested that clarify under which genotype for forage clover the temporal dynamics of plant growth are presented. How do these dynamics differ between ecotypes? Since the ANOVA has been performed, the results of this analysis for each index during various periods should be given, and error bars should be marked in the figure.
7. The experimental design should be expressed in a logical sequence. Begin with the dimensions of the pots and the weight of the soil, followed by the specifics of the experimental design, including the methods for soil moisture treatment and irrigation control. The current description of irrigation control is too vague, and the experimental design treatment is not adequately justified. Particularly, for the treatment of FC26.5%±1.5%, it should be clarified how exceeding the field capacity (FC) is managed, as there could be leakage.
8. Line 329: The notation "50:30:20" is ambiguous. What components, besides sand, are included in the substrate mixture?
9. Is fertilization applied in the experiment, and if so, what and how is it applied?
10. Line 362-364: If the two water treatments involve irrigation when the soil moisture content drops to 25% and 21%, respectively, and the target irrigation levels are 26.5% and 22.5%, then a 1.5% increase in soil moisture would ostensibly require only 0.27L of water for 18kg soil. The origin of the 0.6L of irrigation water mentioned is puzzling in the paper. Additionally, the use of FC to denote field capacity and then as a label for soil moisture treatments is confusing, which should be revised for clarity.
11. All measured variables should be accompanied by specific dates or times of observation as a suggestion.
12. The conclusions should be restructured to include a brief statement of the scientific problem, key findings, and not merely a reiteration of the results. The conclusions should reflect an understanding of the findings, clarify the study's contribution to existing knowledge, and provide a perspective on future research directions.
Comments on the Quality of English Languageno
Author Response
"Please see the attachment."

Reviewer 2 Report
Comments and Suggestions for Authors
The manuscript by Xolocotzi-Acoltzi et al evaluated the growth, productivity, and yield components response of five Lotus corniculatus genotypes to water deficit by pot experiment with seven cut times within one year. The results showed that ecotype 202700 is the best option under water deficit. The study also provided information such as production and yield composition in different growing seasons, tolerance of different genotypes to high or low temperatures, etc. The description of the experimental design, measurement methods, and data analysis are clear. The interpretation of the data the conclusion is appropriate. I therefore would only ask for a few minor revisions.
Line 21-23, this sentence is ambiguous. Modify to two sentences might be better.
Line 25, “The summer and summer-autumn seasons were the ones with the best” The division of seasons is not clearly described in MM. My understanding is seven harvests or cuts. Then the date of each cut should be marked, rather than simply divided into seven seasons.
Line 138 Table 1, The superscript of the letter or not should be uniform. Standard deviation or standard error should be added to the chart. The same applies to other charts.
Line 147, Line 191, Dry stems --> Line 388, dead material
Figure 3,S --> SU or SP, and should be stated in figure legends; Seasons time --> Harvest season or Harvest time
Line 224, Line 349, “L. corniculatus” should be used in italics.
Line 379, PL --> PC
Author Response
"Please see the attachment."
